# Design, Synthesis, In Silico and POM Studies for the Identification of the Pharmacophore Sites of Benzylidene Derivatives

**DOI:** 10.3390/molecules28062613

**Published:** 2023-03-13

**Authors:** Mohammad I. Hosen, Yousef E. Mukhrish, Ahmed Hussain Jawhari, Ismail Celik, Meryem Erol, Emad M. Abdallah, Mohammed Al-Ghorbani, Mohammed Baashen, Faisal A. Almalki, Hamid Laaroussi, Taibi Ben Hadda, Sarkar M. A. Kawsar

**Affiliations:** 1Laboratory of Carbohydrate and Nucleoside Chemistry (LCNC), Department of Chemistry, Faculty of Science, University of Chittagong, Chittagong 4331, Bangladesh; 2Department of Chemistry, Faculty of Science, Jazan University, Jazan 45142, Saudi Arabia; 3Department of Pharmaceutical Chemistry, Faculty of Pharmacy, Erciyes University, Kayseri 38280, Turkey; 4Department of Science Laboratories, College of Science and Arts, Qassim University, Ar Rass 51921, Saudi Arabia; 5Department of Chemistry, Faculty of Science and Arts, Ulla, Taibah University, Medina 41477, Saudi Arabia; 6Department of Chemistry, Science and Humanities College, Shaqra University, Ad-Dawadmi 11911, Saudi Arabia; 7Department of Pharmaceutical Chemistry, Faculty of Pharmacy, Umm Al-Qura University, Makkah 21955, Saudi Arabia; 8Laboratory of Applied Chemistry & Environment, Faculty of Sciences, Mohammed Premier University, Oujda 60000, Morocco

**Keywords:** methyl 4,6-*O*-benzylidene-α-d-glucopyranoside, antimicrobial, PASS prediction, molecular dynamics, POM theory, identification of pharmacophore sites

## Abstract

Due to the uneven distribution of glycosidase enzyme expression across bacteria and fungi, glycoside derivatives of antimicrobial compounds provide prospective and promising antimicrobial materials. Therefore, herein, we report the synthesis and characterization of six novel methyl 4,6-*O*-benzylidene-α-d-glucopyranoside (MBG) derivatives (**2**–**7**). The structures were ascertained using spectroscopic techniques and elemental analyses. Antimicrobial tests (zone of inhibition, MIC and MBC) were carried out to determine their ability to inhibit the growth of different Gram-positive, Gram-negative bacteria and fungi. The highest antibacterial activity was recorded with compounds **4**, **5**, **6** and **7**. The compounds with the most significant antifungal efficacy were **4**, **5**, **6** and **7**. Based on the prediction of activity spectra for substances (PASS), compounds **4** and **7** have promising antimicrobial capacity. Molecular docking studies focused on fungal and bacterial proteins where derivatives **3** and **6** exhibited strong binding affinities. The molecular dynamics study revealed that the complexes formed by these derivatives with the proteins L,D-transpeptidase Ykud and endoglucanase from *Aspergillus niger* remained stable, both over time and in physiological conditions. Structure–activity relationships, including in vitro and in silico results, revealed that the acyl chains [lauroyl-(CH_3_(CH_2_)_10_CO-), cinnamoyl-(C_6_H_5_CH=CHCO-)], in combination with sugar, were found to have the most potential against human and fungal pathogens. Synthetic, antimicrobial and pharmacokinetic studies revealed that MBG derivatives have good potential for antimicrobial activity, developing a therapeutic target for bacteria and fungi. Furthermore, the Petra/Osiris/Molinspiration (POM) study clearly indicated the presence of an important (O1^δ−^----O2^δ−^) antifungal pharmacophore site. This site can also be explored as a potential antiviral moiety.

## 1. Introduction

In biochemistry, carbohydrates are the most active molecules and they are recognized for the many roles they play in the functioning of biological systems. In various biological processes, carbohydrates are involved in viral and bacterial infections, cell growth and proliferation, cell–cell communication, as well as the immune response [1,2]. In addition, they are a source of metabolic energy supply and are involved in the fine-tuning of cell–cell interactions and other biochemical processes [3,4]. In recent years, the dramatic spread of multidrug resistant pathogens has become a worldwide health problem, while since 2010, only a very few new antibiotics have been approved to fight multidrug resistant bacteria. Therefore, there is an urgent need for alternative antibiotics, particularly against Gram-negative bacteria [5]. Healthcare providers are also suffering all over the globe from the same problem with human fungal pathogens such as *Candida* spp. and *Aspergillus* spp. that have developed remarkable resistance to available fungal drugs and are threatening human health and life [6]. Accordingly, the scientific community is striving in a race against time to innovate new antimicrobial drugs from plants [7], nanoparticles [8] and synthesized compounds [9]. In this context, synthesizing new molecules and evaluating their antimicrobial activity constitute essential approaches to discovering new potential antimicrobial and antiviral agents [10,11].

Moreover, many scientific reports claim that the major chemical bioactive molecules are either aromatic, heteroaromatic, or acyl substituent groups [12,13,14,15,16,17,18,19]. In addition, benzene and its derivatives, which contain atoms such as nitrogen, sulphur and halogen are known to improve the biological activity of the parent compound [20,21,22,23]. It has also been reported that the linking of two biologically active nuclei leads to a new molecule—one that might possibly better display biological activity [24,25,26]. Moreover, in several cases of studies carried out in the selective acylation of carbohydrates, it has been noted that the combination of two or more heteroaromatic nuclei with acyl groups contributed to both diversification and the improvement of antimicrobial activity [27,28,29]. In a recent study, some monosaccharide analogs were shown to be suitable inhibitors against protein cancer cells [30,31,32]. Further studies carried out by modifying the nucleoside and monosaccharide structures at the hydroxyl (-OH) group have revealed that the derived molecules are potent anti-SARS-CoV-2 candidates [33,34,35] and antimicrobial agents [36,37].

The present work has two objectives which are mainly the synthesis of a series of derivatives by substituting the hydroxyl group (-OH) of methyl 4,6-*O*-benzylidene-α-d-glucopyranoside (MBG) with various acyls groups (including aromatic and heteroaromatic group), followed by an exploration of the characterization of their physicochemical and biological properties using in vitro and in silico methods. It is important to mention that employing the in silico approach could help to provide insightful information for complicated biological systems [38,39,40]. After establishing the structure and purity of these new derivatives by employing spectroscopic and chromatography methods, their antibacterial and antifungal activities were evaluated in vitro against five collected strains. Density functional theory (DFT) calculations were performed to study the derivatives’ electronic behaviors and geometrical structures. The chemical reactivity indicators, such as HOMO, LUMO and gap energies were also evaluated [41,42]. Molecular docking studies performed on pathogenic fungal and bacterial proteins have helped to obtain more information about the biological applications of synthesized molecules [43,44]. Thus, their binding affinity and abilities with fungal proteins (PDB: 1KS5, 1R51) and bacterial proteins (PDB: 4A1J, 5IQR) were assessed, whereby the binding mode was then identified. To confirm the binding stability of the ligand in the protein, molecular dynamics simulations were undertaken on the docked complexes. Finally, pharmacokinetic prediction was performed to assess and compare the absorption, metabolism and toxicity of MBG derivatives. To identify the nature of the pharmacophore site, a POM study was conducted. There is a difference between antibacterial and antifungal drugs, and the presence of a (X^δ−^----Y^δ−^) moiety would lead to potential antifungal activity more so than an antibacterial (XH^δ−^----Y^δ−^) pharmacophore site. These studies highlight the inhibitory effect of MBG derivatives against bacterial and fungal pathogens using in vitro experiments and in silico computational approaches.

## 2. Results

### 2.1. Chemistry and Characterization

Six newly designed compounds, derivatives of methyl 4,6-*O*-benzylidene-α-d-glucopyranoside (MBG) (from **2** to **7**) were synthesized and characterized. The hydrogen atom of a hydroxyl group in methyl 4,6-*O*-benzylidene-α-d-glucopyranoside was substituted by various acyl chlorides from lauroyl, myristoyl, palmitoyl, triphenylmethyl and cinnamoyl. Figure 1 represents the entire workflow of the current study.

An illustration of the general procedure for synthesizing MBG derivatives is presented in Figure 1.
molecules-28-02613-sch001_Scheme 1Scheme 1General procedure for synthesizing derivatives from **2** to **7** of MBG.
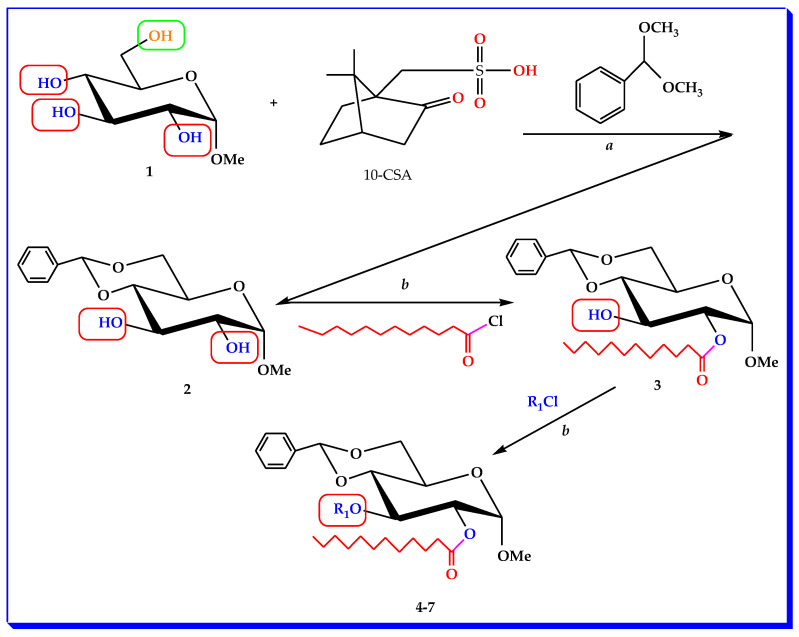

**Compounds****4****5****6****7****R_1_**CH_3_(CH_2_)_12_CO-CH_3_(CH_2_)_14_CO-(C_6_H_5_)_3_C-C_6_H_5_CH=CHCO-

*Reagents and conditions*: *a:* dry DMF, camphor-10-sulphonic acid (10-CSA), benzaldehyde dimethyl acetal, Et_3_N, EtOAc. *b:* anhydrous dichloromethane/Et_3_N, −5 °C, DMAP, stirring for 6 h. R_1_Cl = various acyl chlorides for **4**–**7**.

A preliminary attempt was made to synthesize methyl 4,6-*O*-benzylidene-α-d-glucopyranoside **2** from methyl α-d-glucopyranoside (**1**) by following the Richtmeyer (1962) methodology [45]. Compound **3** was obtained in 83.8% as needles with a melting point of 149–151 °C. In the FTIR spectra (Appendix A Appendix A), absorption bands were observed at 1748 cm^−1^ that resembled carbonyl stretching and at 3401–3510 cm^−1^ (br) that resembled hydroxyl stretching. In its ^1^H-NMR spectrum (Appendix A), two two-proton multiples were found at 2.38 {CH_3_(CH_2_)_9_C*H*_2_CO-} and δ 1.65 {CH_3_(CH_2_)_8_C*H*_2_CH_2_CO-}, respectively. Sixteen-proton multiples at δ 1.29 {CH_3_(C*H*_2_)_8_(CH_2_)_2_CO-} and three-proton multiples at δ 0.90 {C*H*_3_(CH_2_)_10_CO-} were also seen. These findings rationalize the appearance of a lauroyl group in position 2 as depicted by the deshielding of the C-2 proton from its common value (~4.00) [39,40] in the initial diol (**2**) to δ 4.83 (as dd, J = 3.7 Hz, and 9.7 Hz). The ^13^C-NMR spectrum also displayed the appearance of a lauroyl group following the prospective resonance peaks: δ 174.61 {CH_3_(CH_2_)_10_*C*O-}, δ 34.18, 31.94, 29.64 (×2), 29.50, 29.31, 29.30, 29.02, 24.97, 22.73 {CH_3_(*C*H_2_)_10_CO-} and δ 14.17 {*C*H_3_(CH_2_)_10_CO-}. The mass spectra of compound (**3**) had a molecular ion peak at HRMS (ESI, *m*/*z*) [M + H]^+^ 465.5901 that resembled the chemical formula C_26_H_40_O_7_. Plenary exploration of the FT-IR, ^1^H-NMR, ^13^C-NMR and HRMS spectra (Appendix A) led to the confirmation of the structure of compound (**3**) as methyl 4,6-*O*-benzylidine-2-*O*-lauroyl-α-d-glucopyranoside (Appendix A). The FTIR spectra of myristoyl derivative (**4**) (Appendix A) depicted an innate absorption band at 1740 cm^−1^ for carbonyl stretching. In the ^1^H-NMR spectrum, the resonance peaks at δ 2.41 {as 2H, m, CH_3_(CH_2_)_11_C*H*_2_CO-}, 1.64 {as 2H, m, CH_3_(CH_2_)_10_C*H*_2_CH_2_CO-}, 1.27 {as 20H, br, m, CH_3_(C*H*_2_)_10_CH_2_CH_2_CO-} and 0.89 {as 3H, t, J = 6.8 Hz, C*H*_3_(CH_2_)_12_CO-} confirmed the appearance of a myristoyl group in the derivative. The downfield shift of H-3 from its primary value δ 4.22 (as t, J = 9.7 Hz) in the precursor compound **3** to δ 5.62 (as t, J = 9.6 Hz) in compound **4** revealed the existence of a myristoyl group in position 3.

The COSY spectrum of compound **3** started with the signal from the most downfield H-1 protons and was readily assigned. Thus, the H-1 signal at the bottom left of the diagonal showed a cross-peak with H-6a. Hence, a H-1 proton around δ 4.88 was coupled with the hydrogen whose signal was observed around δ 3.96 (i.e., the H-6a proton). Identically, the signal from H-6a was further joined by a cross-peak to the signal from 2H of CH_3_(*CH_2_*)_10_CO- to confirm the coupling between H-6a and 2H, CH_3_(*CH_2_*)_10_CO-. The downfield shift of PhCH-, H-1, H-3, H-4, H-6a and H-6b as compared to precursor compound **3** (Appendix A) clearly indicated the insertion of a lauroyl chain at the C-2 position. The signal appearance determined by analyzing the COSY, HSQC and HMBC spectral experiments (Appendix A), along with the ^13^C-NMR spectrum, confirmed the structure to be methyl 4,6-*O*-benzylidine-2-*O*-lauroyl-α-d-glucopyranoside (**3**).

Further evidence for the structure, according to the 2-*O*-lauroyl derivative (**3**), was gained by the synthesis of palmitoyate (**5**). Using the ^1^H-NMR spectrum, the appearance of the palmitoyl group in the molecule was confirmed by the following resonance peaks: δ 2.38 {2H, m, CH_3_(CH_2_)_13_C*H*_2_CO-}, δ 1.65 {26H, m, CH_3_(C*H*_2_)_13_CH_2_CO-} and δ 0.90 {3H, m, C*H*_3_(CH_2_)_14_CO-}. The insertion of the palmitoyl group in position 3 was clarified by the resonance peaks of C-3 that appeared at δ 5.63 (as t, J = 9.6 Hz), which indicated that this atom’s carbon was noticeably deshielded when compared to its score in the initial compound (**3**). The mass spectra of compound (**5**) had a molecular ion peak at HRMS (ESI, *m*/*z*) [M + H]^+^ 704.0003, resembling the molecular formula C_42_H_70_O_8_. Complete investigations of the spectral data and other parameters assisted in considering the structure of this compound to be methyl 4,6-*O*-benzylidine-2-*O*-lauroyl-3-*O*-palmitoyl-α-d-glucopyranoside (**5**).

The synthesized compound (**6**) was separated in 96.60% yield (189.20 mg) as a crystalline solid with a melting point of 130–133 °C. In its ^1^H-NMR spectrum, six-proton multiples at δ 7.35 (as m, Ar-H)) and nine-proton multiples at δ 7.24 (as m, Ar-H) were revealed as two characteristics for inserting a triphenylmethyl group in the molecule. A cinnamoyl derivative (**7**) was the final MBG derivative prepared. FTIR spectra displayed absorption bands at 1728 cm^−1^ for -CO stretching and 1628 cm^−1^ for -CH=CH- stretching. In the ^1^H-NMR spectrum, two one-proton doublets were found at δ 7.77 (as d, J = 12.1 Hz, PhC*H*=CHCO-) and δ 6.51 (as d, J = 12.1 Hz, PhCH=C*H*CO-), respectively, indicating the presence of a cinnamoyl group in the molecule. In addition, two-proton multiples at δ 7.56 (as m, Ar-H) and three-proton multiples at δ 7.27 (as, m and Ar-H) were due to the one aromatic ring proton. The downfield shift of C-3 to δ 5.60 (as t, J = 9.8 Hz) from its usual value in the precursor compound **3** and the resonances of other protons in their anticipated positions showed the presence of the cinnamoyl group in position 3 of the compound methyl 4,6-*O*-benzylidine-3-*O*-cinnamoyl-2-*O*-lauroyl-α-d-glucopyranoside (**7**) (Appendix A).

### 2.2. Antibacterial Activity

The MBG derivatives (**2**–**7**) were subjected to antibacterial and antifungal screening studies against five human pathogenic bacteria and two phytopathogenic fungi. Compound **7** exhibited the highest zone of inhibition against *S. aureus* (31.0 ± 0.4 mm), followed by *S. abony* (26.5 ± 0.4 mm), *B. subtilis* (21.0 ± 0.3 mm), *P. aeruginosa* (20.0 ± 0.3 mm) and *E. coli* (18.0 ± 0.2 mm). Compound **4** showed a notable zone of inhibition against *S. abony* (24.0 ± 0.3 mm), followed by *B. subtilis* (21.0 ± 0.3 mm), *S. aureus* (16.0 ± 0.2 mm) and *P. aeruginosa* (15.0 ± 0.2 mm), while a weak activity was recorded by *E. coli* (7.0 ± 0.1mm). Inhibition zones of compounds **4** and **7** were exhibited against *B. subtilis* and were higher than those reported by the standard antibiotic azithromycin (19.0 ± 0.2 mm) [46]. As for compounds **5** and **6**, they showed weak or no sensitivity against the tested bacteria compared to the standard azithromycin (Table 1). Based on the diameters of the inhibition zones, the antibacterial activity of the synthesized that showed significant antibacterial activity are **7**, **4**, **5** and **6**. (Appendix A). Our antibacterial result is in harmony with a previous study on different synthesized compounds of α-d-glycopyranosides and β-d-glycopyranosides, which reported that these compounds demonstrated a remarkable broad antibacterial activity against a wide range of bacterial strains (Gram-positive and Gram-negative bacteria) [47].

### 2.3. MIC and MBC Values

The results of the MIC and MBC tests against five bacterial pathogens are shown in Figure 2 and Figure 3 and Appendix A, including the least MIC and MBC values, the highest antibacterial potential with remarkable inhibitory (MIC) and bactericidal (MBC) effects. The tested bacteria showed various inhibitions and susceptibility to the tested compounds (**4** and **7**) that already reported the highest antibacterial activity using a diffusion test. However, based on the MIC and MBC tests, the most susceptible bacteria are *B. subtilis* (MIC: 675.0 ± 0.01 µg/mL, MBC: 2700.0 ± 0.02 µg/mL) and *S. abony* (MIC: 675.0 ± 0.1 µg/mL, MBC: 2700.0 ± 0.02 µg/mL (**7**) and 5400.0 ± 0.04 µg/mL (**4**). The MIC and MBC tests are important for understanding the modes of action and the pharmacodynamics of the chemical compounds [48]. Therefore, further future studies using a wide range of bacterial and fungal strains as well as other pharmaceutical studies are recommended to evaluate the possible utilization of these synthesized derivatives as drugs against specific organisms.

### 2.4. Antifungal Susceptibility

Most of the methyl 4,6-*O*-benzylidene-α-d-glucopyranoside (**1**) derivatives showed good inhibition of mycelial growth (Table 2 and Appendix A) against both of the fungal strains *A. niger* and *A. flavus*. From the derivatives tested, compound **4** inhibited 85.55 ± 0.05% against *A. niger* (Appendix A) and 64.44 ± 0.05% against *A. flavus*. Remarkable mycelial growth inhibition was also observed with derivative **5** against *A. niger* (74.88 ± 0.05) and *A. flavus* (40.00 ± 0.05%). Compound **7** also showed interesting inhibition against *A. niger* (73.33 ± 0.05) and *A. flavus* (38.88 ± 0.05%). In general, the inhibition zones of derivatives **4**, **5**, **6** and **7** were significantly better than those of the standard antibiotic Nystatin. These results are in accordance with our previous publication [49]. The results revealed that the presence of different acyl moieties significantly enhanced the antimicrobial activity of carbohydrates. Moreover, some previous studies on relatively similar synthetic derivatives support the current findings. A series of synthesized methyl α-d-glucopyranoside esters were investigated for their antifungal potential, and the results showed that most of them exhibited more than 70% inhibition of mycelial growth of *A. flavus* [50]. A new derivative of epicatechin glucopyranoside from *Styrax suberifolius* released considerable inhibitory activity against an important pathogenic fungus *Phomopsis cytospore*, with a high rate of inhibition zone (86.72% at 100 μg/mL) [51].

### 2.5. Structure–Activity Relationship

The enhancement of the antibacterial efficacy of the MBG derivatives might be due to the insertion of lauroyl, myristoyl, palmitoyl, trityl and cinnamoyl functional groups. The hydrophobicity of a compound is a vital property when considered with such bioactivity or modification of membrane amalgamation as the membrane permeation indicates. Hunt has also stated that the vigor of aliphatic alcohols must be concerned with their lipid solubility through the hydrophobic interaction between the alkyl group of alcohols’ lipid regions in the membrane. Our findings suggest that a similar hydrophobic interaction might ensue between the acyl chains of MBG accumulated in the lipid similar to the nature of bacteria membranes. Due to this hydrophobic interaction, bacterial pathogens lose their membrane permeability, which ultimately leads to death. Taking this into account, an attempt was made to determine the structure–activity relationship (SAR) of the synthesized compound based on the results. It was observed from the study that the introduction of the lauroyl group at the C-2 position and the myristoyl group at C-3, in addition to the lauroyl group and cinnamoyl group at the C-3 position, plus the lauroyl group at the C-2 position, as with compounds **4** and **7**, enhanced the antimicrobial activity of MBG (**1**) (Figure 4).

### 2.6. PASS Prediction

The antimicrobial spectra using the online server PASS software were predicted for all the MBG derivatives **2**–**7**. The PASS results were expressed in terms of activity probability (Pa) and inactivity probability (Pi), as reported in Table 3. From Table 3, it is evident that the probabilities for MBG derivatives **2**–**7** to be antibacterial, antifungal, antioxidant and anticarcinogenic were different from one another. The probability values were predicted as follows: 0.45 < Pa < 0.58 for antibacterial activity, 0.68 < Pa < 0.76 for antifungal activity, 0.58 < Pa < 0.66 for antioxidant activity and 0.70 < Pa < 0.84 for anticarcinogenic activity. These results show that MBG derivatives are more susceptible to fungi than bacterial pathogens. The linear aliphatic acyls used as substituents increased the antifungal activity (Pa ¼ 0.761) of MGP (1, Pa ¼ 0.521), while the use of ring-bearing and/or branched acyls slightly improved this activity. Compound **4**, in which two linear acyl groups (lauroyl and myristoyl) were present, had the highest probability value for antioxidant activity (Pa ¼ 0.661). In the case of anticarcinogenic activity prediction, the values of probability obtained were such that 0.70 < Pa < 0.84 on all the derivatives. In addition, for any substance, according to the PASS prediction methodology, a value of Pa greater than 0.7 indicates the possibility of detecting, in vitro, the activity under study, as well as perhaps identifying an already existing analogue pharmaceutical agent. It was revealed that the modification of hydroxyl (-OH) groups of MBG significantly increased thermodynamic properties, which indicated the inherent stability of the synthesized derivatives. Moreover, the presence of bulky acylating groups also suggested the possible improvement of these physicochemical properties.

### 2.7. DFT

The larger the HOMO-LUMO energy gap in a molecule, the more stable the molecule (Appendix A). In Appendix A, the stability of the synthesized derivatives is listed as follows: **7** > **3** > **4 > 5 > 2 > 6**. The difference in LUMO energy levels has been predicted to influence the antimicrobial activity of the compounds [52]. The values of the LUMO energy of compounds **4** and **7** were, therefore, in agreement with this prediction. Compounds **4** and **7** had higher antibacterial and antifungal activity; they showed the lowest LUMO energies of −0.68491 eV and −1.95868 eV, respectively. Chemical reactivity indexes [53] such as ionization potential (IP), electron affinity (EA), electronegativity (X), chemical hardness (η) and chemical softness (S) were also calculated for all the synthesized compounds (Appendix A). Molecules with high chemical hardness have little or no intramolecular charge transfer. The low chemical hardness (2.3146 eV) with a high negative value of chemical potential (−4.2732 eV) indicated that derivative **7** consisted of comparatively soft molecules with higher polarizability than the other derivatives in the series. The high electronegativity (4.2732 eV) and electrophilicity (1.9723 eV) of derivative **7** indicated its high power to withdraw electrons, and hence to act as an electrophile.

### 2.8. Biological Validation

In order to identify the ability of molecular docking algorithms to confirm the conformation of protein-bound ligand, redocking of the co-crystallized ligand pose was studied for the validation and effectiveness of the docking results. Appendix A depicts the superimposed pose between the docked ligand conformation and the co-crystallized ligand conformation, and the RMSD value is <2Ǻ. The complex was then observed to interact with all the amino acid residues compared to the similar ones reported in the current study. On the one hand, this large symmetric structure can interact in the active site of a protein during docking, as is the case in this study; hence, the RMSD score would be at a very high level. On the other hand, the small molecules can attain a low RMSD score easily, even when imposed randomly. The outcomes of this visual observation showed the same interactions as in the experimental binding mode, as found in docking. This result means that the RMSD score alone is unreliable or not dependable data for the validity of docking poses; therefore, visual inspection of new data is essential.

### 2.9. Molecular Docking

Based on the results of both the antibacterial and antifungal activity of the compounds, molecular docking studies were performed on two fungal proteins and two bacterial proteins. The calculations of iGEMDOCK binding affinities on the 1KS5, 1R51, 4A1J and 5IQR proteins are given in Appendix A. As shown in Appendix A, the lowest binding affinity for the compounds with 1KS5 and 1R51 proteins was consistently observed with derivatives **4**, **5**, **6** and **7**, while the highest was observed with derivatives **2** and **3**, regardless of the target protein. The lower the binding affinity, the stronger the biological activity. Under these conditions, the evolution sequence observed in the case of the target protein, 1KS5, remained in conformity with that observed in the case of the fungal strain *Aspergillus niger*. Regarding bacterial activity, the binding affinity values predicted the effects of all compounds against the *Bcillus subtilis* target proteins, whereby the highest binding affinity was observed with compound **5**, unlike the experimental results that showed only compounds **4** and **7** as truly active. Nevertheless, in all cases, the biological activities of compounds **4** and **7** demonstrated by the in vitro tests were well predicted by the in silico methods. All these results were examined for further information by visualizing the protein–ligand complexes formed by compounds **4** and **7** with the 4A1J and 1KS5 proteins as 2D and 3D (Figure 5A–F and Appendix A). On the 4A1J protein, compound **3** formed hydrogen bonds with TYR 110 (4.25A), ASN 198 (4.78 Å), ASN A: 63 (4.43 Å) and GLN A: 200 (5.77 Å), while compound **7** formed hydrogen bonds with SER A: 16 (4.60 Å). ILE A: 197 (4.97 Å and 5.57 Å) and SER A: 111 (4.72 Å and 4.33 Å). It was confirmed via the 3D diagram that compounds **4** and **7** showed similar binding modes at the active site. On the 1KS5 protein, compound **4** formed hydrogen bonds with SER A: 14 (3.98 Å), while derivative **7** formed hydrogen bonds with SER A: 136 (4.26 Å) and CYS A: 139 (3.68 Å).

In the case of these two proteins, the binding energy of compound **4** was lower than that of compound **7**. Analysis of the established hydrogen bonds revealed that the binding affinity of the compounds depends on both the number of residues involved and the interaction distances (Appendix A).

### 2.10. Molecular Dynamic Simulations

A molecular dynamics (MD) simulation process was performed with a time period of 100 ns to analyze the structural behavior of the protein–ligand complexes formed with L,D-transpeptidase Ykud from *B. subtilis* and endoglucanase from *A. niger* under physiological conditions [54]. The analysis of the stability of these complexes was also performed over time. An ideal protein–ligand complex would fluctuate less and would interact with the ligand in MD simulations when an inhibitor was fixed at the active site of the target. In general, RMSD values that are less than 0.3 mm are considered to be suitable for protein stability, and the lower this value, the more stable the complex. As seen in Figure 6A, the RMSD values for both complexes increased rapidly in the first 10 ns, with the 1KS5-**4** complex up to 0.25 nm and the 1KS5-**7** complex up to 0.3 nm. The 1KS5-**7** complex stabilized between 10 and 50 ns at around the interaction distance of 0.2 nm. After 50 ns, slight fluctuations were observed until the highest RMSD value was reached, at 70 ns. The complex stabilized again at a distance of 0.25 nm between 90 and 100 ns. The same phenomena were observed with the 1KS5-**4** complex (Figure 6A), which stabilized between 20 and 50 ns, at around 0.3 nm. After 50 ns, the complex fluctuated slightly and reached its maximum RMSD value at 70 ns. It stabilized again at 0.35 nm between 90 and 100 ns. In particular, the complex structure formed by the 1KS5-**7** complex (Figure 6B) with endoglucanase A from *A. niger* was more stable and had lower RMSD values than the complex structure formed by the 1KS5-**3** complex. In Figure 6C, it can be observed that for the 4A1J-**4** and 4A1J-**7** complexes, rapid fluctuations occur in the first 10 ns, rising up to 0.4 nm, and then remaining stable at 0.2 nm. Only the 4A1J-**7** complex showed slight fluctuations at 70 and 90 ns, up to 0.3 nm. The average RMSD values of the 1KS5-**4**, 1KS5-**7**, 4A1J-**4** and 4A1J-**7** complexes were determined at 0.271 nm, 0.214 nm, 0.199 nm and 0.195 nm, respectively. It was observed that all of the complexes formed stable and consistent structures. To assess the flexibility of the complexes, the fluctuations of each amino acid residue shown by the RMSF graphs were plotted. A high RMSF value (peaks) indicated the presence of loops, turns, terminal ends and loose bonds, indicating the flexibility of the protein structure, while a lower value indicated the presence of secondary structures such as sheets and helices that provide stability to the structure. As illustrated in Figure 6B, RMSF values were determined below 0.8 nm for the 1KS5-**4** complex and below 0.5 nm for the 1KS5-**7** complex, with both complexes showing the highest fluctuation for the 25th and 160th residues. The 1KS5-**4** complex showed a higher RMSF value than the 1KS5-**7** complex, indicating higher flexibility and lower stability of the 1KS5-**4** complex as compared to the 1KS5-**7** complex. As shown in Figure 6D, the 4A1J-**4** complex (0.5 nm) showed a higher RMSF value than the 4A1J-**7** complex (0.4 nm), indicating higher stability of the 4A1J-**7** complex. Therefore, it is expected that it will be able to establish a strong bond with these proteins. The complexes also both showed the highest fluctuations for the 25th and 160th residues.

### 2.11. MM/PBSA

The molecular mechanics/Poisson–Boltzmann surface approach (MM/PBSA) is one of the most widely used methods in the study of biomolecular complexes for the calculation of interaction energies. Free energy calculations were only related to the following protein–ligand complexes: 1KS5-**4**, 1KS5-**7**, 4A1J-**4** and 4A1J-**7**. The values obtained are reported in Table 4. From Table 4, the binding free energies of the protein–ligand complexes are: −133.254 ± 20.879, −156.093 ± 23.130, −162.362 ± 22.027 and −112.863 ± 30.839 kJ/mol, respectively. All of these energies are negative, thus reflecting the spontaneous formation and stability of the complexes. However, Van der Waals interactions contributed better to the stabilization and formation of complexes, since the corresponding energies were the most negative. However, the energies of the polar solvation interactions were positive, indicating the effects of destabilization due to polarization. Of all the complexes studied, compound **7** exhibited the lowest binding free energies regardless of the protein. Therefore, it was expected that it would be able to establish a strong bond with these proteins.

### 2.12. Pharmacokinetics Properties

The prediction of pharmacokinetics properties is of great importance in pharmacology, toxicology and pharmacokinetics, especially for drug compound candidates. Different parameters have been defined for a rational approach in understanding these processes. It has been reported that the candidate molecule must satisfy the following rules: MW ≤ 500 (g/mol), MLogP ≤ 4.15, HBD ≤ 10 and HBA ≤ 5 [55]. These rules are defined as follows: 160 ≤ MW ≤ 480, −0.4 ≤ WLogP ≤ 5.6, 40 ≤ MR ≤ 130 and 20 ≤ atoms ≤ 70. The TPSA value is an indicator of the substance ease of use by the body [42]. Molecules with a TPSA greater than 140 Å^2^ are believed to have a low capacity for penetrating cell membranes, while those with TPSA ≤ 60 Å^2^ are easily absorbed [56] (Appendix A). If a high TPSA value accounts for poor penetration of molecules in a hydrophobic environment such as biological membranes, it may account for their ready penetration in hydrophilic environments such as the core of transporter proteins. Appendix A shows the state of MBG derivatives and compliance with these rules following the calculated values of descriptor parameters. Compounds **4**, **5** and **6** did not meet Lipinski’s rules and, furthermore, had a low rate of gastrointestinal absorption.

### 2.13. POM Analyses: Identification of the Pharmacophore Sites

Pharmacophore sites can be identified, one by one, on the basis of their different physico-chemical parameters and the different electronic charge repartition of corresponding heteroatoms. This new theory of POM analyses has been extended, with success, to other diverse and different biotargets. Here, the goal was to identify the pharmacophore sites of the series of compounds **2**–**7**. Therefore, the identification of the type of pharmacophore sites of these compounds was derived from the physical and chemical properties of the tested compounds by using the Petra/Osiris/Molinspiration (POM) analyses platform. The results of the pharmacokinetic properties and bioactivity score analysis are shown in Table 5 and Table 6. The Osiris analysis of the series of tested compounds **2**–**7** shows that most of the compounds have no side effects, except compound **7**, which is an irritant (Table 5). The Molinspiration analysis of the series of compounds tested clearly shows that there are more substituents than necessary because the molecular weight is more than the limit (MW = 500 g/mole). This constitutes the first violation of Lipinski’s five rules. A second violation appeared when the cLogP value was greater than 5 (Table 5, Table 6 and Appendix A).

The atomic charge calculations of heteroatoms of compounds **2**–**7** show that when two oxygen atoms that are negatively charged constitute the terminal atoms of a pocket, there is net bioactivity (Figure 7 and Figure 8). The POM study indicates the presence of an important (O1----O2) antifungal pharmacophore site. In this promising investigation, more drug-likeness in vitro and in vivo studies such as nontoxic concentration toward healthy cells may be conducted in the forthcoming manuscript.

## 3. Materials and Methods

### 3.1. General Information

The solvents used were of analytical grade and purified using standard procedures. Thin-layer chromatography (TLC) was performed on silica gel GF_254_ (Kieselgel/MilliporeSigma; Darmstadt, Germany). Melting points (m.p.) were determined using an electrothermal m.p. apparatus. Infrared spectral analyses were performed using a Fourier-transform infrared (FTIR) spectrophotometer (IR Prestige-21, Shimadzu, Japan). ^1^H-NMR and ^13^C-NMR spectra were recorded using a Brucker Avance DPX-400 MHz. Mass spectra of synthesized compounds were obtained by liquid chromatography/electrospray ionization tandem mass spectrometry (LC/ESI-MS/MS, abbreviated as LC-MS for convenience). Column chromatography was performed using silica gel G_60_. All reagents used are commercially available (MilliporeSigma). Evaporations were performed under reduced pressure on a Buchi rotary evaporator (MilliporeSigma, Germany).

### 3.2. Synthesis of MBG Derivatives

#### 3.2.1. Methyl 4,6-*O*-benzylidene-α-d-glucopyranoside (**2**)

A solution of methyl-α-d-glucopyranoside (**1**) (5 gm, 25.74 mmol) in dry dimethylformamide (DMF) (3 mL) with stirring was treated with benzaldehyde dimethylacetal (5 mL, 33.5 mmol) and a catalytic amount of camphor-10-sulphonic acid (100 mg), and the mixture was heated at −5 °C for 6 h. After cooling to room temperature, the mixture was neutralized with Et_3_N, diluted with EtOAc, washed with saturated NaHCO_3_ and brine, and dried over Na_2_SO_4_. The progress of the reaction was monitored by TLC (ethyl acetate/hexane, 3:1), and then the solvent was removed. The residue was purified via passage through a silica gel column with ethyl acetate/hexane (3:1) as an eluant to afford methyl 4,6-*O*-benzylidene-α-d-glucopyranoside (**2**) (5.5 gm, 81%) as a white crystalline solid. This compound was sufficiently pure for its use as the starting material for the acylation reactions, as reported in this dissertation. The structure of this compound (m.p. = 163–164 °C) was ascertained by analyzing and comparing its FTIR, ^1^H-NMR and ^13^C-NMR spectra with the reported data [45].

#### 3.2.2. Methyl 4,6-*O*-benzylidine-2-*O*-lauroyl-α-d-glucopyranoside (**3**)

To a stirred solution of methyl 4,6-*O*-benzylidene-α-d-glucopyranoside (**2**) (100 mg, 0.35 mmol) in anydrous dichloromethane (3.5 mL)/Et_3_N at −5 °C, lauroyl chloride (0.087 mL, 1.1 molar eq.) was added, and stirring continued at 0 °C for 6 h. The reaction mixture was then allowed to stand at room temperature overnight. The progress of the reaction was monitored by TLC (CH_3_OH/CHCl_3_, 1:24), which indicated full conversion of the starting material into a single product (R*_f_* = 0.52). The syrup was passed through a silica gel column and eluted with CH_3_OH/CHCl_3_ (1:24) provided to the lauroyl derivative (**3**). The compound was sufficiently pure for use in the next stage without further purification or identification.

White needles, 83.8% yield, m.p. 149–151 °C; FTIR (KBr) ν/cm^−1^ 1748 (C=O), 3401–3510 (br, -OH); HRMS (ESI, *m*/*z*) [M + H]^+^ 465.5901; Analysis calcd for C_26_H_40_O_7_: C, 67.22, H, 8.67%; found: C, 67.23, H, 8.24%.

^1^H-NMR and 1H-NMR data are presented in Table 7 and Table 8.

#### 3.2.3. General Procedure for the Synthesis of 2-*O*-lauroyl Derivatives (**4**–**7**)

Myristoyl chloride (0.31 mL, 3.5 molar eq.) was added to a cooled solution of the 2-*O*-lauroyl compound (**3**) (153 mg, 0.33 mmol) in anhydrous dichloromethane (4 mL)/Et_3_N, and the solution was stirred at room temperature for 6 h. Conventional work-up, as described earlier for compound **3**, followed by chromatographic purification (CH_3_OH/CHCl_3_, 1:24, as eluant), gave the myristoyate **4** (212.50 mg) (R*_f_* = 0.53). A similar reaction and purification procedure was applied to prepare compound **5** (palmitoyl derivative, 192.20 mg), compound **6** (triphenylmethyl derivative, 189.20 mg) and compound **7** (cinnamoyl derivative, 190.15 mg).

#### 3.2.4. Methyl 4,6-*O*-benzylidine-2-*O*-lauroyl-3-*O*-myristoyl-α-d-glucopyranoside (**4**)

White needles, 95.82% yield, m.p. 139–140 °C; FTIR (KBr) ν/cm^-1^ 1740 (C=O); HRMS (ESI, *m*/*z*) [M + H]^+^ 675.9512; Analysis calcd for C_40_H_66_O_8_: C, 71.18 and H, 9.85%; found: C, 71.17 and H, 9.86%.

#### 3.2.5. Methyl 4,6-*O*-benzylidine-2-*O*-lauroyl-3-*O*-palmitoyl-α-d-glucopyranoside (**5**)

White needles, 96.45% yield, m.p. 147–148 °C; FTIR (KBr) ν/cm^−1^ 1744 (C=O); HRMS (ESI, *m*/*z*) [M + H]^+^ 704.0003; Analysis calcd for C_42_H_70_O_8_: C, 71.76 and H, 10.03%; found: C, 71.78 and H, 10.04%.

#### 3.2.6. Methyl 4,6-*O*-benzylidine-2-*O*-lauroyl-3-*O*-(triphenylmethyl)-α-d-glucopyranoside (**6**)

White crystalline solid, 96.60% yield, m.p. 130–133 °C; FTIR (KBr) ν/cm^−1^ 1735 (C=O); HRMS (ESI, *m*/*z*) [M + H]^+^ 707.9101; Analysis calcd for C_45_H_54_O_7_: C, 76.46 and H, 7.69%; found: C, 76.45 and H, 7.67%.

#### 3.2.7. Methyl 4,6-*O*-benzylidine-3-*O*-cinnamoyl-2-*O*-lauroyl-α-d-glucopyranoside (**7**)

White needles, 97.29% yield, m.p. 165–166 °C; FTIR (KBr) ν/cm^−1^ 1728 (C=O); HRMS (ESI, *m*/*z*) [M + H]^+^ 595.7312; Analysis calcd for C_35_H_46_O_8_: C, 70.68 and H, 7.79%; found: C, 70.70 and H, 7.81%.

### 3.3. Microbial Strains

The referenced ATCC and NCTC bacterial and fungal strains were generously provided by the Department of Microbiology at the University of Chittagong. Seven pathogenic microbes including five bacteria and two fungi were used in the antimicrobial investigations (Appendix A). The provided bacterial strains were subcultured in nutrient broth (Oxoid, Hampshire, UK) and incubated for up to 18 h at 30–35 °C to reach the exponential phase, where the fungal strains were subcultured in potato dextrose agar (Oxoid, UK) at 25 °C for up to three days. Bacterial strains were adjusted to the 0.5 McFarland standard prior to use to obtain a bacterial suspension containing about 1.5 × 10^8^ colony-forming units/mL. A fresh fungal sample was taken directly from the subcultured plates.

### 3.4. In Vitro Antibacterial Activity

The antibacterial activities of the MBG derivatives (**2**–**7**) (in 5% DMSO) at a concentration of 21 mg/mL were determined against the five bacterial strains by the disc diffusion method [57]. Four mm diameter paper discs were saturated with the MBG derivatives (one disc holds about 10 µL). The discs were then loaded onto a set of plates containing Mueller–Hinton agar streaked with the tested bacterial strains. The positive control was azithromycin 15 μg/disc (BEXIMCO, Bangladesh Ltd., Dhaka, Bangladesh) and the negative control was discs saturated with 5% DMSO. All plates were incubated for 24 h at 35–37 °C. After incubation, the possible zone of inhibition was measured in MM. The test was repeated three times and the mean was calculated.

### 3.5. In Vitro Antifungal Activity

The antifungal activities of the produced MBG derivatives (**2**–**7**) were examined using the poisoned food technique [58]. In brief, 20 mL of potato dextrose agar was placed onto sterile petri plates, and a determined quantity of the tested compounds (21 mg/mL in 5% DMSO) was added and mixed well, and then allowed to solidify. As a negative control, another plate containing the same agar medium supplemented with 5% DMSO was tested. Under aseptic conditions, fungal disks (5 mm in diameter) from an 8-day-old pure culture were put in the middle of a Petri plate containing medium and cultured at 27 to 28 °C for seven days. After a week of incubation, the growth of each fungus was inspected and documented. The percentage of inhibition (mycelial growth inhibition) was calculated using the following formula:$$\%\;Inhibition=\frac{100\;(Control-Treatment)}{Control}$$

### 3.6. MIC and MBC Tests

The minimum inhibitory concentration (MIC) was determined using the microdilution technique, as recommended by the Clinical and Laboratory Standards Institute (CLSI) with some modifications [59]. In a sterile 96-well series, two-fold dilutions were made for the tested compounds in descending concentrations, and at the end, 100 µL was present in each well. The diluent was 5% DMSO, which was also examined and served as a negative control; 95 µL of sterile Mueller–Hinton broth was loaded into each well, and 5 µL of the adjusted bacterial suspension was added. The plates were incubated for 18 h at 35–37 °C. An indicator of 10 µL 2,3,5-triphenyltetrazolium chloride 0.5% (*w*/*v*) solution (Sigma-Aldrich, Munich, Germany) was added to each well to detect the microbial growth in the wells. The MIC was recorded when no microbial growth was observed. After determining the MIC of the derivatives, the content of each well was seeded on a Mueller–Hinton agar plate in order to determine the minimum bactericidal concentration (MBC). Among the different wells, one is treated without using chemicals, serving as a reference for negative control, while another is treated with azithromycin (standard antibiotic), serving as a reference for positive control. The MBC was determined by finding the lowest concentration at which 99.9% of the initial bacterial inoculum is eliminated.

### 3.7. Structure–Activity Relationship

A structure–activity relationship (SAR) analysis was employed to predict the antimicrobial activity from the molecular structure of a pharmaceutical target. This well-known technology is often used in drug design processes to guide the acquisition or synthesis of new compounds with desirable properties. In the present study, the SAR study was analyzed according to the Hunt [60] and Kim [61] membrane permeation concepts.

### 3.8. Quantum Chemical Calculations

The quantum parameters of the overall reactivity, in particular, were determined in the ground state of each MBG derivative. In this context, the geometry optimization and frequency calculations were carried out with the Gaussian09 software [62] using the DFT method through its functional B3LYP and the basis of functions 6-311G (d,p) implemented in the program. The parameters determined were the HOMO-LUMO energies, ionization potential (IP), electron affinity (EA), electronegativity (χ), chemical hardness (η), chemical softness (S), chemical potential (μ) and electrophilic index (ω). The graphical interface GaussView6.0.16 [63] was used to visualize the molecules’ structures.

### 3.9. Prediction of the Activity Spectra

The properties of the biologically active derivatives were described using the online PASS (prediction of activity spectra for substances) software developed by Lagunin et al. [64]. The predicted results are expressed in terms of probabilities of biological activity (Pa) or inactivity (Pi), as the activity is predicted by comparing the structure of a new compound with the structure of a well-known biological structure. Different values of probability are followed by comments and interpretation, such as (i) if Pa > 0.7, the substance is very likely to exhibit activity in the experiment and the chance of this substance being the analogue of a known pharmaceutical agent is high; (ii) if 0.5 < Pa < 0.7, the substance is likely to exhibit activity in an experiment with less probability and the substance is unlike known pharmaceutical agents, and (iii) if Pa < 0.5, the substance is unlikely to exhibit the activity during an experiment.

### 3.10. Biological Validation

The biological validation of docking outcomes was investigated by extracting the co-crystallized ligand (8-azaxanthine) of the fungal protein (PDB ID: 1R51), and re-docked while maintaining the same position. The ground state energy pose attained on re-docking and the co-crystallized ligands were superimposed, and its root mean square deviation (RMSD) score was calculated between these two superimposed ligand poses. To validate the docking procedure, the RMSD score must be within the reliable range of 2 Å [65].

### 3.11. Molecular Docking

The molecular docking studies were performed against endoglucanase A from *Aspergillus niger* and L,D-transpeptidase Ykud from *B. subtilis* using the iGEMDOCK version 2.1 software [66]. Their crystal structures were obtained from the RSCB protein data bank website with the code PDB ID: 1KS5 (2.10 Å), 1R51 (1.75 Å), 4A1J (2.20 Å), 5IQR (3.00 Å). The 3D structures of the minimized ligands were prepared using the ChemDraw3D 19.0 software. Standard docking accuracy settings were employed for the initiation of interactions with ligands and binding sites. Chimera V.1.15 and Discovery Studio Visualizer v2021 were used to demonstrate interactions between ligands and proteins. Then, the results were analyzed.

### 3.12. Molecular Dynamic Simulations

The stability of the protein–ligand complexes was studied according to the molecular dynamic simulations carried out using the online version of the GroMACS 2019.2 software (GROningen MAchine for Chemical Simulations) [67]. The topology parameters for all the derivatives were generated using the online GlycoBioChem PRODRG2 server [68] (http://davapc1.bioch.dundee.ac.uk/cgi-bin/prodrg (accessed on 1 December 2022) when the structures of 1KS5 and 4A1J for the simulation process were subjected to a GROMOS 43A1 force field [69] and solvated with water following the SCP model. These phases were carried out for a time period of 0.3 ns under the pressure of 1 atm, at a temperature of 300 K according to the V-rescale thermostat [70] and using the Parrinello–Rahman barostat conditions [71]. Root mean square deviations (RMSDs) and root mean square fluctuations (RMSFs) were also determined. The free binding energy was calculated between 80 and 100 ns using the tool g_mmpbsa developed by Rashmi Kumari et al. [72], which implements the molecular mechanics Poisson–Boltzmann surface area (MM-PBSA) approach.

### 3.13. Prediction of the Pharmacokinetic and ADME Properties

A SwissADME program package is an online tool for computing physicochemical descriptors as well as predicting ADME parameters, pharmacokinetic properties, drug-like nature and medicinal chemistry friendliness of one or multiple small molecules [73]. In this work, parameters such as TPSA value, molecular weight, hydrogen bond acceptor and donor atom number, lipophilicity, absorption status in the gastrointestinal tract, factors of Lipinski’s and Ghose’s [74,75] rules were evaluated and analyzed.

### 3.14. POM Analyses

The theory of POM (Petra/Osiris/Molinspiration) analyses was developed by a group at Taibi Ben Hadda in collaboration with the American NCI and TAACF and has led to real success in pharmacology and the drug design field. By using the theory of POM (Petra/Osiris/Molinspiration) analyses, it is now easier to identify and optimize most of the antibacterial [76], antifungal [77], antiviral [78], antiparasital [79] and antitumor [80] pharmacophore sites, one by one, on the basis of the different physico-chemical parameters and their different electronic charge repartition of corresponding heteroatoms. Moreover, the theory of POM analyses has been extended, with success, to other diverse and different biotargets [81].

## 4. Future Perspectives

Based on the literature review and investigated data, we may conclude that this study plays a vital role in the field of drug development. Carbohydrate derivatives, especially those having aromaticity, are potent for antimicrobial treatment. Researchers across the world are committed to discovering more efficacious and protected antimicrobial drug candidates to medicate diseases originated by pathogenic microorganisms. In these circumstances, the synthesis of novel carbohydrate analogues and evaluation of their antimicrobial efficacy is the best way to establish effective antimicrobial agents. Our synthesized compounds are novel with a high percentage of yield, and their structures were confirmed by spectral analysis. Then, the PASS prediction, POM analyses and in vitro experimental study showed logical alignment in the case of the antimicrobial activity investigation. This observation was rationalized by an in silico computational approach. Molecular docking results also exhibited the high antimicrobial potential of the synthesized compounds through strong binding affinity and interaction with fungal and bacterial proteins. Furthermore, the docked complex was found stable in molecular dynamics. In addition, SAR analyses showed the strong efficacy of the compounds to inhibit microorganisms. Finally, the ADMET study revealed safe scores for most of the compounds. The whole investigation clarified a very preliminary part of the drug development process. Undoubtedly, it needs more wet-lab experiments and an analytical approach to consider our compounds as potent drug candidates.

## 5. Conclusions

Synthesis, spectral analysis, physicochemical, antimicrobial, molecular docking, molecular dynamics and ADMET studies of MBG and its derivatives were conducted. The characteristics made it possible to highlight the biological and chemical reactivity behavior of the newly synthesized compounds. In addition, the various biological tests carried out revealed that compound **4** and compound **7** both exhibited strong and effective inhibitory activity against some bacterial and fungal organisms. The use of aliphatic and aromatic groups as substituents was favorable for the improvement of biological activity in the MBG series. The determination of the reactivity indicators by the DFT method showed that the MBG derivatives **3**–**5** had a HOMO-LUMO gap greater than that of MBG, thus indicating their greater electronic stability with respect to the parent molecule (MBG). The molecular docking and dynamic studies coupled with those of ADMET prediction made it possible to rationalize all of these observations and to reveal the promising potential of MBG derivatives as candidates for the achievement of effective antimicrobial activity. Most of the derivatives showed remarkable binding affinity and binding ability, in particular, when analyzing interactions with endoglucanase A from *A. niger* and L,D-transpeptidase Ykud from *B. subtilis*. The protein–ligand complexes are predicted to be highly stable in a biological system. The prediction study of drug-likeness properties established that the derivatives displayed acceptable pharmacokinetic properties and a spectrum of various biological activities. Furthermore, the POM study indicated the presence of an (O1----O2) moiety that constituted a potential antifungal pharmacophore site. This (O1----O2) pharmacophore site should also be explored as an antiviral site. This depends only on distance (O1---O2). All methods, tools and techniques used in this work formed a solid basis for the results obtained. Therefore, particular attention paid to these results and these molecules will undoubtedly lead to the development of antimicrobial treatments against pathogenic microbes such as *A. niger* and *B. subtilis*.

## Data Availability

Data are available in this article and Appendix A.

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
