# Peer review of "Design, Synthesis, In Silico and POM Studies for the Identification of the Pharmacophore Sites of Benzylidene Derivatives"

_molecules, 2023, doi:10.3390/molecules28062613_

Round 1

Reviewer 1 Report

The research paper has reported the synthesis and characterization of six novel methyl-4,6-O-benzylidene-α-D-glucopyranoside (MBG) derivatives. Spectroscopic techniques and elemental analysis has been carried out to validate the structures of newly synthesized derivatives. Anti-bacterial and anti-fungal potential of these novel derivatives has also been assessed by exposing them to microbial and fungal strains and by executing MIC and MBC tests. In addition, DFT, PASS prediction, POM analysis, molecular dynamic simulations, molecular docking, pharmacokinetic properties have been also implemented to provide complete insight of novel derivatives.

There are some observations which need to be taken care of:

Generally, the references are appropriate. However there are few spacing issues in references in authors names and numbers. Full stop is missing and year name is not bold at few places. These should be checked again and changed accordingly.

·         Formatting of Scheme 1 should be improved. Almost all structures need to be cleaned up. Structure numbering has not been applied on each number. Use of similar type of arrows would give better outlook.

·         Figure S4 and S6: Numbers should be added in horizontal manner not erratically placed.

·         Clean up of structures is required in figure 4.

·         Clean up structures in Table S2, figure S2 and figure S3 .

·         In abstract, numbers should come in chronological order i.e. 4, 5, 6 and 7, instead of 7, 4,5, and 6.

·         Justification of reaction conditions is required under scheme 1. All lines should be in accordance.

·         Line 189, ‘While’, w should be in small letter.

·         Line 493, 0 oC should be written with space.

·         Line 450, ‘ligand,GPCRL,’ insert space.

·         Carefully bold all structure numbers in methods and materials section.

·         Line 589, correct the superscript of degree sign. It should be 25 oC.

·         Line 600, inset space and accurate degree sign as 35-37 oC.

·         Line 607, insert space ‘5%DMSO’.

·         Line 610 and 623, correct superscript of degree sign.

·         Line 642, correct spacing between d and p.

·         Line 645, remove one extra parenthesis from w)).

·         Heading of molecular docking, shift it to next page to improve outlook.

·         Line 701, remove extra space between ‘now  easier’

Also, provide spectra of all the compounds. 

Author Response

Responses of Comments

Reviewer-1

  1. Generally, the references are appropriate. However there are few spacing issues in references in authors names and numbers. Full stop is missing and year name is not bold at few places.

Respond: Thank you very much. We have corrected according to your suggestions.

  1. Formatting of Scheme 1 should be improved. Almost all structures need to be cleaned up. Structure numbering has not been applied on each number.

Respond: We have modified Scheme 1.

  1. Figure S4 and S6: Numbers should be added in horizontal manner not erratically placed.

Respond: Corections have been made.

  1. Clean up of structures is required in figure 4.

Respond: We have modified figure 4.

  1. Clean up structures in Table S2, figure S2 and figure S3.

Respond: Corrections have been done.

  1. In abstract, numbers should come in chronological order i.e. 4, 5, 6 and 7, instead of 7, 4,5, and 6.

Respond: Corrections have been done.

  1. Justification of reaction conditions is required under scheme 1. All lines should be in accordance.

Respond: Corrections have been done.

  1. Respond: We have corrected the following points in the manuscript:
  • Line 189, ‘While’, w should be in small letter.
  • Line 493, 0 oC should be written with space.
  • Line 450, ‘ligand,GPCRL,’ insert space.
  • Carefully bold all structure numbers in methods and materials section.
  • Line 589, correct the superscript of degree sign. It should be 25 oC.
  • Line 600, inset space and accurate degree sign as 35-37 oC.
  • Line 607, insert space ‘5%DMSO’.
  • Line 610 and 623, correct superscript of degree sign.
  • Line 642, correct spacing between d and p.
  • Line 645, remove one extra parenthesis from w)).
  • Heading of molecular docking, shift it to next page to improve outlook.
  • Line 701, remove extra space between ‘now  easier’

  1. Also, provide spectra of all the compounds. 

Respond: Spectra were provided in the Supplementary file.

Reviewer 2 Report

This manuscript describes synthesis of benzylidene derivatives as well as their molecular docking, dynamics, ADMET studies of MBG, and antimicrobial performance. However, the way of paper presentation and some cases lacks information on the compounds that need to be addressed. The manuscript needs improvement and modify all parts accordingly and some of the issues are attached below.

Concerns to author

  1. Title need attention to reader, and I think it is recommended to change as a shorter.
  2. Scheme 1 should be rearranged as they are not logically presented and include all in details of reactions.
  3. I recommend authors include NMR analysis in tabular form instead long paragraphs in main article.
  4. I strongly encourage authors to provide spectra of all derivatives with NMR window range 1—10 ppm/0-210 ppm for 1H/13C and rearrange Si file properly.
  5. I can see that most of the cases manuscript riddled with typos as well as space issues between units and values and please recheck these minors for further improvement.

Author Response

Responses of Comments

Reviewer-2

  1. Title need attention to reader, and I think it is recommended to change as a shorter.

Respond: Thank you very much. We have shorter the Title according to your suggestion.

  1. Scheme 1 should be rearranged as they are not logically presented and include all in details of reactions.

Respond: We have modified Scheme 1.

  1. I recommend authors include NMR analysis in tabular form instead long paragraphs in main article.

Respond: We have presented 1H- and 13C NMR data of the compounds 3-7 in tabular form (Table 7 and Table 8).

  1. I strongly encourage authors to provide spectra of all derivatives with NMR window range 1—10 ppm/0-210 ppm for 1H/13C and rearrange Si file properly.

Respond: We have added in the SI file.

  1. I can see that most of the cases manuscript riddled with typos as well as space issues between units and values and please recheck these minors for further improvement.

Respond: We checked space.

Round 2

Reviewer 1 Report

The paper has been modified and can now be accepted subject to fulfillment of journal's formatting requirements.

Reviewer 2 Report

The authors have made significant improvements in the present version and can be accepted for publication.